# Sickle Cell Anemia and *Babesia* Infection

**DOI:** 10.3390/pathogens10111435

**Published:** 2021-11-04

**Authors:** Divya Beri, Manpreet Singh, Marilis Rodriguez, Karina Yazdanbakhsh, Cheryl Ann Lobo

**Affiliations:** 1Department of Blood Borne Parasites, Lindsley F. Kimball Research Institute, New York Blood Center, New York, NY 100065, USA; dberi@nybc.org (D.B.); msingh@nybc.org (M.S.); mrodriguez@nybc.org (M.R.); 2Department of Complement Biology, Lindsley F. Kimball Research Institute, New York Blood Center, New York, NY 100065, USA; kyazdanbakhsh@nybc.org

**Keywords:** *Babesia*, sickle-cell anemia, hemolysis, haemoglobinopathies

## Abstract

*Babesia* is an intraerythrocytic, obligate Apicomplexan parasite that has, in the last century, been implicated in human infections via zoonosis and is now widespread, especially in parts of the USA and Europe. It is naturally transmitted by the bite of a tick, but transfused blood from infected donors has also proven to be a major source of transmission. When infected, most humans are clinically asymptomatic, but the parasite can prove to be lethal when it infects immunocompromised individuals. Hemolysis and anemia are two common symptoms that accompany many infectious diseases, and this is particularly true of parasitic diseases that target red cells. Clinically, this becomes an acute problem for subjects who are prone to hemolysis and depend on frequent transfusions, like patients with sickle cell anemia or thalassemia. Little is known about *Babesia*’s pathogenesis in these hemoglobinopathies, and most parallels are drawn from its evolutionarily related *Plasmodium* parasite which shares the same environmental niche, the RBCs, in the human host. In vitro as well as in vivo *Babesia*-infected mouse sickle cell disease (SCD) models support the inhibition of intra-erythrocytic parasite proliferation, but mechanisms driving the protection of such hemoglobinopathies against infection are not fully studied. This review provides an overview of our current knowledge of *Babesia* infection and hemoglobinopathies, focusing on possible mechanisms behind this parasite resistance and the clinical repercussions faced by *Babesia*-infected human hosts harboring mutations in their globin gene.

## 1. Introduction

Human babesiosis is a zoonotic disease in which the natural acquisition of human cases is most often the result of an interaction with established zoonotic cycles [1,2]. A number of factors have contributed to the emergence of human babesiosis, including increased awareness among physicians, changing ecology, and an increased population of immuno-compromised individuals who exhibit severe disease. *Babesia* belongs to the Phylum Alveolata, Class Apicomplexa Family Piroplasmida and Genus *Babesia* which comprises more than 100 classified species. The four identified *Babesia* species that can infect humans are: *B. microti*, *B. divergens*, *B. duncani* and *B. venatorum*. As molecular techniques are becoming more available and accessible, new species described as “*microti*-like” or “*divergens*-like” are being described [3].

*Babesia* is an intra-erythrocytic parasite that causes malaria-like symptoms in infected people. *Plasmodium*, the causative agent of malaria, is the most studied Apicomplexan parasite and, like *Babesia*, resides within red blood cells. *Plasmodium* has a long association with its human host dating back to the first report in 1857 [4]. As the erythrocyte provides the parasite with the infra-structure to grow and multiply, it is expected that any perturbation to the cell should impact parasite homeostasis and viability. Clinical, epidemiological, and genome-wide association studies have identified multiple polymorphisms in the globin protein of hemoglobin within the red blood cell (RBC), commonly referred to as hemoglobinopathies, that attenuate or completely abrogate malaria pathogenesis. Malaria has thus imposed extreme selective pressure on the human genome, far more than any other infectious disease, and the RBC has been the prime target for evolutionary adaptation. The evolutionary proximity of *Plasmodium* and *Babesia* [5], and the fact that they both infect RBCs, raises important clinical questions of *Babesia* infections in patients harboring hemoglobinopathies.

In this paper, we review the literature documenting the effects of hemoglobinopathies on the life cycle of the *Babesia* parasite, using both in vitro and in vivo models of *Babesia* infection. We provide an overview of available clinical cases of the severity of *Babesia* infection in patients harboring these mutations and emphasize why it is essential to focus research in this area. We also describe plausible mechanisms that could exert this protective effect and discuss ways we can use this double-edged sword to develop better therapeutics against blood-borne parasites.

## 2. Pathogenesis and Anemia in Babesiosis

Hemolytic anemia is the central feature of sickle cell anemia (SCA) that contributes to its severe clinical outcomes. Epidemiological studies and basic research point to the pathogenic role of intravascular hemolysis as the primary cause of clinical complications in SCA. Interestingly, the primary pathological event in babesiosis is also hemolysis, resulting in hemolytic anemia and jaundice. In the absence of aggressive intervention, the anoxia and toxic effects that follow often lead to organ failure and death. Parasitemias do not always relate directly to the degree of anemia, suggesting that erythrocyte destruction is not only due to lysis of infected cells or their removal by splenic and liver macrophages, but also due to lysis of bystander cells which might be a significant contributing factor to the process. Some symptoms, such as fever, myalgia, renal insufficiency, coagulopathy, and hypotension, that occur in *B. microti* infections with parasitemias of less than 1%, may be caused by excessive production of pro-inflammatory cytokines, as also seen in malaria [6,7].

Clinical features in heavy infections, particularly those caused by another major human species, *B. divergens*-like parasites occurring in immuno-compromised patients, exhibit acute illness that appears suddenly with hemoglobinuria as a presenting symptom [8,9]. The clinical presentation also includes persistent non-periodic high fever (40–41 °C), shaking chills, intense sweats, headaches, myalgia, and lumbar and abdominal pain. Jaundice may develop as a result of the high level of hemolysis; vomiting and diarrhea may be present, and the toxins and anoxia, resulting from the hemolysis and the host immunological response, may cause respiratory, cardiac, renal, or hepatic failure [10,11,12]. The few known infections with *B. venatorum* have shown similar though generally milder manifestations [12].

In our previous report of *B. microti* infections in mice, we reported the increase in hemolysis in *Babesia*-infected mice, which was highly accentuated in mice harboring the SCA genotype, as observed by significantly reduced hematocrit and enhanced hemoglobinuria in these mice [13]. Therefore, these studies indicate that hemolysis is a central mechanism of clinical manifestations of both babesiosis and SCA individually, which is further accentuated and becomes life-threatening in *Babesia* infections in SCA mice/humans.

We found in our infected SCA mouse model that they mount an equally robust adaptive immune response despite exhibiting low parasitemia. This underscores the importance of examining both the fate of *B microti* and the immunological consequences of parasite infection in individuals with SCA to establish whether a similar hyperimmune response against the parasite occurs in humans too. Patients with SCA require transfusions, with some undergoing chronic transfusion therapy, placing them at greater risk of acquiring transfusion-transmitted infections like babesiosis. Thus, these individuals, if transfused from an infected donor, would be exposed to a larger infectious dose compared with a tick bite. The outcome of these infections, whether one of immune protection mediated by the first infection or a more deleterious pathological sequel, is required to be studied to establish effective treatments for these patients [13].

Laboratory findings that are consistent with mild-to-moderate hemolytic anemia include a low hematocrit, low hemoglobin level, low haptoglobin level, elevated reticulocyte count, and elevated lactate dehydrogenase level [14]. Thrombocytopenia is commonly observed. The illness usually lasts for 1 or 2 weeks, but fatigue may persist for months. Asymptomatic parasitemia may persist for several months after standard therapy is initiated or for more than a year if the patient does not receive treatment. Illness may relapse in severely immuno-compromised patients despite 7 to 10 days of antimicrobial therapy and may persist for more than a year if not adequately treated [15].

Patients infected by *B. microti* show a wider range of signs and symptoms. A study on Block Island, Rhode Island, USA, concluded that about 25% of adults and 50% of children are asymptomatic or only show very mild ‘flu-like’ symptoms in cases that may not result in medical consultation and are therefore rarely diagnosed [16]. At the other end of the spectrum, very severe manifestations, similar to those seen in *B. divergens* infections, may occur in patients who have been splenectomized, are receiving immune-suppressive therapy, or are elderly. These cases typically show high fever, chills, night sweats, myalgia, hemolytic anemia, and hemoglobinuria [17]. Life-threatening complications include acute respiratory failure, disseminated intra-vascular coagulation, congestive heart failure, coma, and renal failure [18]. Immuno-compromised individuals are also likely to develop persistent relapsing disease despite treatment [15]. The symptoms caused by *B. duncani* and related parasites (CA1–4) closely resemble those of *B. microti* infections [19].

## 3. *Babesia* and the Red Blood Cell

When *Babesia* sporozoites are first injected into the human host, they target the host RBCs immediately, unlike *Plasmodium* spp. which are required to undergo an exo-erythrocytic phase in hepatic cells. Furthermore, *Babesia*-infected RBCs remain circulating in the peripheral blood stream, including regularly passing through the hosts’ spleen, and do not sequester to the fine capillaries of the bone marrow or organs. It is the parasite’s ability to first recognize and then invade host RBCs that is central to human babesiosis, and the parasites invade RBCs using multiple complex interactions between parasite proteins and the host cell surface, which are not fully elucidated yet [20]. Once inside the RBC, the parasite begins a cycle of maturation and growth exhibited by intense intra-cellular proliferation leading to populations described as 1N, 2N, 4N and >4N [21]. The parasite population can expand inside the RBCs or egress at multiple points in the life cycle [21]. Previous work from our lab has led to the development of synchronized parasite populations and showed the sequential progression of the seven morphological forms of *B. divergens* in culture along with the dynamics of parasite proliferation and differentiation. These processes are maintained through controls that secure the constituent infected-RBC populations in strict ratios to enable rapid movement between new invasion events or further intra-RBC development and replication cycles, as dictated by the environment of the parasite. The early stages of the cycle are morphologically indistinguishable from *Plasmodium* spp., with both appearing as ring-like parasites. However, unlike *Plasmodium*, *Babesia* exhibits plasticity in its life cycle and is thus able to swiftly respond to environmental conditions like host RBCs and nutritional availabilities [22].

## 4. Hemoglobinopathies

For intracellular parasites, the environment of host red cells plays a key role in the development and success of the pathogen; therefore, perturbations in the RBCs are most likely to modify parasite survival and viability. For blood-borne parasites like *Plasmodium* and *Babesia*, the environments of the RBC, membrane proteins, and hemoglobin, the primary oxygen carrier, are important determinants of parasite success. Human hemoglobin is comprised of alpha and beta globin chains encoded from multiple globin genes. The α-globin gene cluster is at the end of chromosome 16 and contains three genes. The human β-globin gene cluster consists of five genes arranged in chromosome 11 in the same order in which they are expressed during human development: ɛ, Gγ-, Aγ-, δ-, and β-globin gene. The Hb switching event which occurs after birth in the β-globin cluster leads to the suppression of the γ-globin gene accompanied by the complementary increase of the previously silent β-globin gene. Understanding the regulation of Hb switching can have direct therapeutic applications for sickle cell disease in which the γ-globin gene can functionally substitute for mutations in the β-globin gene of these diseases [23]. Hemoglobinopathies are genetic disorders of the globin protein and are classified as structural hemoglobin variants including HbS, HbC and HbE, as described ahead, or thalassemia syndromes [24]. The term ‘thalassaemias’ collectively refers to several different genetic mutations that result in either reduced or absent expression of one or more of these globin alleles. Specifically, individuals described as having ‘α-thalassaemia’ have a loss of one or more α-globin allele(s). Additionally, there is also HbH disease (loss of 3 α-globin alleles) and, finally, hydrops fetalis (loss of all 4 α-globin alleles), which leads to the death of the fetus in the uterus. Individuals with mutations in HBB can also have a range of genetic defects referred to as ‘β-thalassaemia’, including β-thalassaemia minor (reduced expression of one β-globin allele), and β-thalassaemia major (reduced expression of both β-globin alleles) [25].

Sickle cell disease (SCD) is the most common monogenic blood disorder of hemoglobin synthesis, encompassing the single replacement mutation of glutamic acid at position 6 of the β-globin chain by valine (HbSS genotype) [26,27,28]. The hallmark of SCD is “the sickle-shaped” red blood cells due to the polymerization of mutated sickle hemoglobin (HbS) under low oxygen tension. Chronic blood transfusion is one of the most effective treatments in SCD and results in the reduction of the frequency of acute pain episodes and acute chest syndrome but causes a dramatic increase in the risk of transfusion-transmitted infection [29]. The HbSS and HbAS (heterozygous) genotypes are commonly found in populations from sub-Saharan Africa. The Hemoglobin C (HbC) mutation (HbAC–heterozygotes; HbCC–homozygotes) also involves a point mutation at the 6th codon in the HBB gene, resulting in a glutamic acid to lysine substitution and is most common in West Africa, with prevalence reported as high as 15% in parts of Burkina Faso [30]. The Hemoglobin E (HbE) mutation is a point mutation that results in a glutamic acid to lysine switch at position 26 of the HBB gene and is most commonly found in parts of Southeast Asia and India and reaches a prevalence of up to 60% in some areas [31,32]. HbS, HbC, and HbE are characterized as structural hemoglobin variants. The major human hemoglobinopathies and related genetic mutations are summarized in Table 1. According to CDC Reports in 2010, the total incidence estimate for sickle cell trait was 15.5 cases per 1000 births in USA, ranging from 0.8 cases per 1000 births in Montana to 34.1 cases per 1000 births in Mississippi. The U.S. incidence estimate for sickle cell trait (based on information provided by 13 states) was 73.1 cases per 1000 black newborns, 3.0 cases per 1000 white newborns, and 2.2 cases per 1000 Asian or Pacific Islander newborns. The incidence estimate for Hispanic ethnicity was 6.9 cases per 1000 Hispanic newborns. The total number of babies born with sickle cell trait in 2010 was estimated to be greater than 60,000. The study showed that as many as 1.5% of babies born in the United States have the sickle cell trait [33]. With approximately 7% of the worldwide population being carriers, hemoglobinopathies are the most common monogenic diseases and one of the world’s major health problems. This makes it very essential to understand the pathogenesis of blood-borne parasites in human hosts harboring these mutations in their RBCs.

## 5. Natural Resistance against Blood-Borne Parasites

The long association and co-evolution of the malaria parasite with humans is reflected in the fact that almost all examples of molecular evolution in humans, like sickle cell anemia, G6PD-deficiency, and thalassemia, are attributed to a selection of mutations that attenuate malaria pathogenesis. Though these mutations lead to unpleasant consequences, as of 2015 it was estimated that about 4.4 million people have sickle cell disease, while an additional 43 million have sickle cell traits [36]. Zones that are endemic for malaria have a high proportion of humans carrying these mutations either in the homozygous form (the subject suffers from the disease caused due to the mutation) or heterozygous form (one copy of normal gene and one copy of mutated gene). These genes have all arisen in areas in which falciparum malaria is endemic, and their rise to high levels of prevalence is thought to result from their conferring significant degrees of protection against this dreaded pathogen. It is well-established that the homozygotes suffer from sickling of RBCs but do not support the rapid growth of the parasite; however, in rare cases, subjects with SCD and malaria can suffer from hyper-hemolytic crisis [37]. Heterozygotes do not suffer from sickling and have lesser severity of malaria. AS subjects can get malaria, but the number of parasitized cells is low, and they rarely suffer from cerebral malaria or severe anemia [37]. The enhanced resistance of persons with sickle traits to falciparum malaria is substantial. Infected AS children have lower parasite densities than AA children and are 50–90% less likely to progress to a severe form of malaria or to die from the disease [38]. *Babesia*’s association with hemoglobinopathies is not completely understood but is an important field of research as hemolytic anemia is common in hemoglobinopathies and can be life-threatening when coupled with *Babesia* infection.

There have been several explanations as to why β-globin might confer resistance to malaria. Researchers have reported that parasitized AS cells sickle more readily and show enhanced HbS polymerization under hypoxic conditions and are therefore removed from circulation. Further, it has been shown that parasites are fragile and killed by these HbS polymers. A compelling cause of reduced parasite load in AS and SS RBCs is the extent of oxidative damage which is inherent in these host cells added to the oxidative stress due to parasite growth; the cumulative oxidant damage can cause considerable damage to the host RBC and impairment of parasite development [39]. Interestingly, accumulated reactive oxygen species (ROS)-mediated damage is a common mechanism shared by AS, SS, G6PD-deficient, β- and α-thalassemia RBCs in mediating resistance to malaria [40,41]. However, the mechanism for ROS-mediated protection in malaria remains elusive. It was also observed that AS RBCs parasitized with *P. falciparum* late stages bound to human microvascular endothelial cells and blood monocytes half as effectively as did comparably infected AA RBCs. Moreover, infected AS RBCs displayed slightly reduced and highly uneven distribution of expression of PfEMP-1 on their surface. There is also evidence based on host microRNAs playing a role in protection in AS and SS RBCs [38,42,43,44,45,46,47,48,49,50]. There have been several plausible mechanisms proposed for resistance of hemoglobinopathic RBCs to malaria, but little is known about *Babesia* in this regard. Given the parallels between the two parasites, it is tempting to speculate that they might share mechanisms of resistance to growth in hemoglobinopathic RBCs.

## 6. *Babesia* and the Sickle Red Cell

The RBC serves as the home for this intra-erythrocytic parasite for its entire life cycle in the human host. The interactions between the parasite and the RBC can be classified into three broad areas: invasion, growth and maturation within the RBC, and egress. Previous studies from our group have examined these phases of the intra-erythrocytic lifecycle of *Babesia divergens* in homozygous SS and heterozygous AS human blood [51]. While the invasion was similar across all RBCs, there was atypical population progression, a potential loss of merozoite infectivity, and defective egress of the parasite in SS cells (as explained in Figure 1). Unlike previous reports in *Plasmodium*, AS cells supported invasion, growth, and egress of *Babesia* much like AA cells. While parasites grew from their characteristic 1N to 2N, 4N and >4N populations in AA and AS cells, in SS host cells beyond 24 h, the majority of the parasites were stuck in the 1N phase as demonstrated in Figure 1. Interestingly, even when parasites growing in SS RBCs were supplemented with fresh AA RBCs, they did not grow [51]. This indicates that the initial invasion and growth in SS host cells programs the parasite irreversibly to poor growth and/or defects in egress. Our work on the mouse model using *B. microti* also showed poor growth of the parasite in mice harboring the SS gene (HbSS-Townes mice), and normal growth in AS mice when compared to the wildtype AA mice [13]. For AA and AS mice, parasitemias peaked on day 7 of infection, while the SS mice exhibited a sluggish increase in parasitemias. In all three genotypes, parasites were cleared by day 21 and all mice survived. Interestingly, while the parasitemia was 4–5-fold lower in SS mice, the extent of immune response mounted was the same in AA, AS and SS mice. The adaptive immune response was measured by a robust GC reaction and significant expansion of TFH cells. Currently, it is not known how these SS mice respond to subsequent *Babesia* infection. This becomes especially critical to understand as babesiosis is primarily a transfusion-transmitted infection and since several sickle cell patients undergo repeated blood transfusions, they may be exposed to a parasite load that could be more than the parasites in one tick bite.

CDC reported that from 1979 to 2009, 159 transfusion-related *Babesia microti* cases were identified, most (77%) of which were from 2000 to 2009 [52]. A recent review in 2016 has summarized the state-wise seropositivity of *B. microti* in the blood used for transfusion in the USA [53]. However, no data are available on the number of sickle patients transfused with *Babesia*-contaminated blood.

## 7. Clinical SCA and Babesiosis

There have been scattered case reports of babesiosis in sickle cell patients transmitted via blood transfusions. Transfusion-acquired babesiosis can result in severe hemolytic anemia in patients with sickle cell disease. The infection can be difficult to treat and may require a prolonged treatment duration [54]. A recent study presented a case of two sickle cell patients who had delayed diagnosis post transfusion due to confusing symptoms: as patients who receive chronic transfusions are also at risk for the development of allo- and autoantibodies, the hemolytic anemia caused by the former can often obscure a different pathophysiology, such as babesiosis, which occurred with these patients. In another case, diagnosis took 4 months for a patient with HbSS and babesiosis, after repeated visits to the hospital [54,55]. Another study reported a young female with HbSC who presented in the emergency department multiple times with pain and shortness of breath, eventually developing unresponsiveness and a brief episode of pulseless electrical activity. She was admitted to the intensive care unit with multisystem organ failure and found to have diffuse ischemic strokes. Infectious workup revealed disseminated anaplasmosis and babesiosis, which had likely caused sickle cell crisis. The patient continued to show a significant neurologic burden, despite months of treatment [56]. Evidently, with increased awareness about babesiosis among physicians and more sensitive diagnostic tests available, the number of case reports on babesiosis have increased. Given that there is a significant incidence of SCA carriers/other hemoglobinopathies in *Babesia*-endemic zones (like north-eastern USA) who require frequent transfusions, it becomes increasingly necessary to understand *Babesia* pathogenesis in such subjects.

## 8. Plausible Mechanisms for Resistance of *Babesia* in Sickle Cells

From previous work of our group, the outstanding question is why the parasite exhibits developmental/egress abnormalities when growing in a sickle cell as opposed to a wild type AA RBC host. There are multiple possible causes for this: It is well known that sickle cells have a high burden of oxidative stress due to repeated polymerization and depolymerization of hemoglobin. It is also well known that a wide variety of intracellular pathogens like *Plasmodium*, *Mycobacterium,* and several viruses impose redox stress on their host cells. Therefore, unfavorable host cell conditions, like increased hemoglobin autoxidation, accumulation of iron in membranes, increased membrane damage, and a shorter red cell life span, could justify the reason why SS cells do not promote growth of the *Babesia* species. Further, the bystander effect, whereby uninfected RBCs are also affected leading to increased hemolysis, has been widely described in malaria. It is possible that in SS subjects there is an accentuated bystander effect leading to massive hemolysis and therefore unfavorable conditions for the parasite to grow and proliferate. It is also possible that invasion/growth of the parasite modifies the shape of the sickling RBCs, making them more prone to splenic removal. Our study in the mouse model has clearly shown that even though parasitemias are much lower in SS mice as compared to AA mice, the adaptive immune response is almost as severe; therefore, the heightened immune response against the parasite might be another SS-specific strategy to abrogate the growth of *Babesia*. In future studies, it would be interesting to monitor the growth of *Babesia* in other hemoglobinopathic disorders like thalassemia and RBC enzymopathy like G6PD-deficiency, both of which are known to afford resistance to malaria. However, unlike *Plasmodium*, AS mice showed no protection against babesiosis and followed the same parasite growth curve both in vitro and in vivo. Thus, from the current experiments, heterozygotes for the mutated beta globin gene do not seem to be protected against babesiosis.

## 9. Concluding Remarks and Future Directions

Hemoglobin-associated genetic disorders affect millions throughout the world and are concentrated in humans living in malaria-endemic countries. However, as borders of countries are becoming more porous, these genetic traits are now seen throughout the world. For several years, researchers have observed that hemoglobinopathies afford protection from malaria and the studies from our group in *Babesia* also point in that direction. Given the evolutionary proximity between these two parasites, it is possible that resistance to their growth in SS cells has a common mechanism. Further studies are needed to understand if the growth of *Babesia* parasites in thalassemic RBCs and those with an inherent deficiency in the G6PD enzyme is similarly impaired and to determine how these mutations hinder intra-erythrocytic parasite growth. These results will provide researchers with an opportunity to discover the Achilles’ heel of two deadly parasites and learn how nature has evolved a way to protect against these diseases. Uncovering the mechanism behind this protection will lead us to a better understanding of their pathogenesis as well as in designing better drugs against these parasites. As described above, multiple mechanisms of resistance against parasite proliferation in sickle cells may operate to confer protection. A detailed study of these pathways is needed to identify the main pathways in *Babesia*-infected red cells and this, in turn, will shed light on the intricate interplay between polymorphisms of the human host red cells and intruding parasites.

## Figures and Tables

**Figure 1 pathogens-10-01435-f001:**
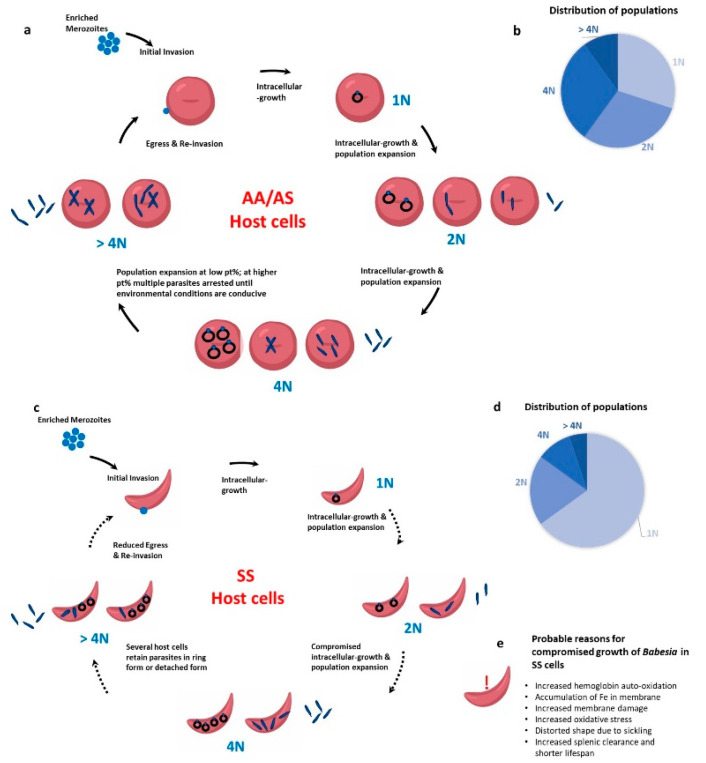
*Babesia* infection progression in wild type (AA) RBCs, heterozygous for sickle cell anemia (AS) RBCs, and homozygous for sickle cell anemia (SS) RBCs. (**a**) In AA and AS host cells, the merozoite invades and the parasite develops inside the RBCs to 1N, 2N, 4N and >4N populations. Egress can take place at 2N, 4N or >4N stage. (**b**) The distribution percentage of 1N, 2N and 4N parasites is similar. (**c**) In SS host RBCs, parasites mostly retain their “ring form” and very few “Maltese cross” forms are seen. (**d**) As shown in the pie chart, a high population of parasites get stuck in the 1N form. (**e**) List of probable reasons for compromised growth of *Babesia* parasites in SS cells.

**Table 1 pathogens-10-01435-t001:** Major hemoglobinopathies and related genetic mutations.

Hemoglobinopathy	Mutation	Position
HbS	Glutamic Acid to Valine	β-6
HbC	Glutamic Acid to Lysine	β-6
HbE	Glutamic Acid to Lysine	β-26
**Thalassemia**	**Gene Modifications**	**Disease Name**
α-Thalassemia	- - / - α	HbH disease
	- - / - -	α-Thalassemia major
β-Thalassemia	β°/β	β-Thalassemia minor
	β°/β°	β-Thalassemia major

Denotes: (-) loss of α-globin gene. (β°) loss of β-globin [34,35].

## Data Availability

Not applicable.

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
