# Peer review of "Sickle Cell Anemia and Babesia Infection"

_pathogens, 2021, doi:10.3390/pathogens10111435_

Round 1

Reviewer 1 Report

In this review manuscript, Beri et al. provide an overview of the current understanding of several aspects of the babesia parasite infection pathogenesis, mainly focused on the hemoglobinopathy clinical settings.

The MS is well written and thoughtfully discusses the most relevant topics of babesiosis.

Minor comments:

1) "Human hemoglobin is comprised of two α-globin and two β-globin proteins and are encoded by two α- globin genes (HBA1 and HBA2) and one β-globin gene (HBB)." ( lines 148-150).
This statement could be refined a bit better for the sake of accuracy: the human (and the mouse) alpha and beta-globin loci contains more than just "two" alpha and "one" beta genes" (indeed, in the beta-globin locus, there are at least five distinct beta-like encoding  genes, with some fetal forms also possibly expressed by adult RBC (especially in the context of globinopathies...)

Author Response

Response: We thank the Reviewer for the remarks. As suggested, we have modified the information on alpha and beta gene loci and added the reference associated with it (Lines 145-150)

Reviewer 2 Report

In this manuscript, Beri and colleagues address the question of sickle disease on parasite growth and disease outcome of Babesia infection in cultures, mice and humans. While hemoglobinopathies have significant impact of malaria pathogenesis in endemic areas, their role in human babesiosis is poorly studied. From this aspect, this review article is an important contribution. The manuscript is well written and addresses a wide-ranging aspects of sickle cell disease.

Specific comments:

  1. A table showing amino acid substations in major hemoglobinopathies in humans would be used for the general reader.
  2. Please describe the course of B. microti infection and pathogenesis in the SCA mice compared to the wild-type mice.
  3. Are the receptors in Sickle Red Cell distinct from those from those in normal RBC utilized for B. diverges invasion? Also, is it possible to maintain  long-term B. divergens culture in sickle red cells.
  4. Would it be possible to comment on severity of microti disease in sickle cell patients?
  5. A population level prevalence of sickle cell disease in different ethnic groups in US and risk of transfusion--transmitted infections would be useful.

Author Response

We thank the reviewer for providing constructive feedback on our manuscript.

  1. A table showing amino acid substations in major hemoglobinopathies in humans would be used for the general reader.

Response: The table has been added with reference in the manuscript in Lines 178-180.

  1. Please describe the course of B. microti infection and pathogenesis in the SCA mice compared to the wild-type mice.

Response: This has been previously described and has been now elaborated in in Lines 238-250

  1. Are the receptors in Sickle Red Cell distinct from those from those in normal RBC utilized for B. diverges invasion? Also, is it possible to maintain  long-term B. divergens culture in sickle red cells.

Response: The invasion machinery of Babesia in sickle cells has not been investigated. In our experiments we see a similar and sometimes a higher invasion in SS cells compared to AA RBCs.

We have successfully maintained B. divergens culture in sickle red cells for 6 days (with no new addition of blood). However, it may be worthwhile trying to propagate them in sickle red cells long-term (the major bottleneck would be continuous supply of sickle red cells in this case).

  1. Would it be possible to comment on severity of microti disease in sickle cell patients?

Response: Line 250-270 explain the currently available data of babesiosis in sickle cell patients. Unfortunately, very few clinical reports are available on babesiosis in SCA subjects.

  1. A population level prevalence of sickle cell disease in different ethnic groups in US and risk of transfusion--transmitted infections would be useful.

Response: Lines 175-180 provide statistics of subjects suffering from different hemoglobinopathies (HbS, HbC and HbE). Additonal information has been added in Lines 180-188.

Statistics on risks of transfusion transmission in USA have been added in Line numbers 258-262.